# Bone Marrow Fibrosis Grading Using Attention-Based Multiple Instance Learning

**Lauren M. Zuromski**[1] (iD)                    LAUREN.ZUROMSKI@ARUPLAB.COM
[1] *Institute for Research and Innovation, ARUP Laboratories, Salt Lake City, UT*

**Alexandra E. Rangel**[1] (iD)                    ALEXANDRA.RANGEL@ARUPLAB.COM
**Muir J. Morrison**[1] (iD)                       MUIR.MORRISON@ARUPLAB.COM
**Paul M. English**[1] (iD)                        PAUL.ENGLISH@ARUPLAB.COM
**Nicholas C. Spies**[1,2] (iD)                    NICK.SPIES@ARUPLAB.COM
**Brendan O'Fallon**[1] (iD)                       BRENDAN.O'FALLON@ARUPLAB.COM
[2] *Department of Pathology, University of Utah, Salt Lake City, UT*

**David P. Ng**[1,2] (iD)                          DAVID.NG@ARUPLAB.COM

**Editors:** Accepted for publication at MIDL 2025

## Abstract

An attention-based multiple instance learning approach is used to improve bone marrow fibrosis (BMF) grading from whole slide images of bone marrow core biopsies. Slide-level labels were parsed from biopsy reports using a large language model, and features were extracted using our fine-tuned DINOv2-based backbone. The model achieved good agreement between BMF predictions and labels ($R^2 = 0.72$, $\kappa = 0.58$). Attention maps showed the model focused on diagnostic regions, highlighting its accuracy and interpretability.

**Keywords:** Bone marrow fibrosis, multiple instance learning, whole slide imaging, digital pathology, large language model, DINOv2

## 1. Introduction

Bone marrow fibrosis (BMF) is a condition where reticulin fibers accumulate, thickening the marrow and impairing blood cell production. BMF is a hallmark of myelofibrosis (MF), a rare myeloproliferative neoplasm (MPN) with prognosis influenced by fibrosis grade, age, genetic mutations, and risk of progression to leukemia. Accurate grading is essential for assessing severity and guiding treatment. BMF is commonly graded on an integer scale from 0 to 3, ranging from no to extensive reticulin fiber intersections (Thiele et al., 2005). Grading relies on qualitative assessments by pathologists examining bone marrow (BM) core biopsy whole slide images (WSIs), assigning the highest grade present in $\geq 30\%$ of the marrow. Although there is general agreement among experts (Kvasnicka et al., 2016), the BMF grading scale poorly handles fibrosis heterogeneity (Ryou et al., 2022).

To address this, a recent study estimated BMF on a continuous scale by modeling ranks of fibrosis severity, improving MPN subtype differentiation and risk stratification (Ryou et al., 2022). However, this method required expert review of thousands of tiles pairs, and the slide-level BMF score was simply calculated as the average of the tile-level grades.

In our ongoing study, we refine fibrosis grading through an attention-based multiple instance learning (ABMIL) approach, where the model learns image tiles that are important

using only slide-level labels. Features were derived using our fine-tuned backbone, and slide-level labels were extracted from biopsy reports using a large language model (LLM). Our model highlights diagnostically relevant regions and shows strong performance, suggesting improved objectivity and predictive power over prior methods.

## 2. Methods

For this analysis, we used clinical WSIs and core biopsy reports spanning 2019–2024 from our reference laboratory. The WSIs were digitized from slides, and the biopsy reports were retrieved from internal databases.

BMF grades were extracted from biopsy reports using Ollama, an open-source implementation of the Llama 3.1 70B LLM (Touvron et al., 2023; Grattafiori et al., 2024). Fibrosis grades were extracted from report text, with heterogeneous descriptions (e.g., "MF-2 to focally MF-3") assigned intermediate values (e.g., 2.5) to align with discrete labels.

WSIs of reticulin-stained BM core biopsies for each BMF-labeled case were tiled to generate the evaluation dataset. An object detection model isolated core tissue to guide recursive tiling from low to high magnification, removing background via color filtering. The evaluation dataset included 322 training and 80 testing cases, each using 512 tiles ($224 \times 224$ pixels at 40x optical objective) that are down- or upsampled as needed.

We used an ABMIL framework, allowing the model to learn from image tile sets with only slide-level labels (Ilse et al., 2018). Features were extracted using a ViT-L/14 backbone fine-tuned with a DINOv2-based approach (Oquab et al., 2023), trained on millions of core tiles from various stains and magnifications, and passed to the ABMIL framework. DINOv2 leverages self-supervised learning through vision transformers, capturing pathology-specific features without labels. For benchmarking, we used common histopathology backbones: Virchow2 (Zimmermann et al., 2024), UNI (Chen et al., 2024), and Prov-GigaPath (Xu et al., 2024). Model optimization used a batch size of 16, learning rate of 0.00025, and early stopping based on lowest test-set RMSE.

## 3. Results and Discussion

The ABMIL model using our in-house backbone performed comparably to the benchmarks, achieving an RMSE of 0.52 (Table 1). Since the in-house backbone is intended for production use, it was necessary to demonstrate performance comparable to widely used histopathology backbones. Ongoing efforts of scanning more cases to expand the training sets for both the ABMIL model and backbone could help improve model performance.

We evaluated the ABMIL model trained with our in-house backbone on both continuous and discrete fibrosis grade scales. There is a strong correlation between predicted and actual values ($R^2$=0.72, $p < 0.0001$, Figure 1). For categorical analysis, we excluded intermediate labels and rounded predictions, yielding 67 cases. Predictions deviate from actual grades by one grade at most (Figure 1). Cohen's Kappa was 0.58 [0.42–0.73], indicating moderate agreement between predictions and labels, which exceeds expert inter-rater variability ($\kappa$=0.51 [0.48–0.54]) (Kvasnicka et al., 2016).

ABMIL models offer interpretability via attention maps, highlighting regions influencing predictions. In a correctly predicted grade-three case, the model correctly focuses on regions

with extensive fibrosis while down-weighting bone and adjacent tissue, which are areas known to skew severity estimates (Figure 2).

Table 1: Model performance (RMSE [95% CI]) across different backbones.

| Backbone | In-house | UNI | Virchow2 | Prov-GigaPath |
|---|---|---|---|---|
| RMSE [95% CI] | 0.52 [0.45–0.61] | 0.53 [0.46–0.63] | 0.54 [0.47–0.64] | 0.54 [0.47–0.65] |

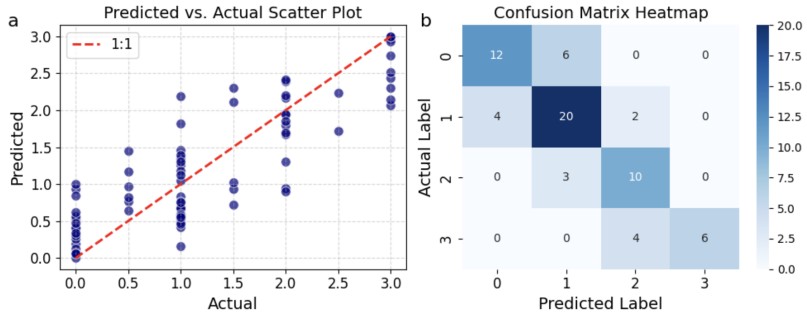

Figure 1: a) Predicted vs. actual labels using the in-house backbone ($R^2$=0.72, $p < 0.0001$). b) Confusion matrix showing agreement between actual and rounded predictions.

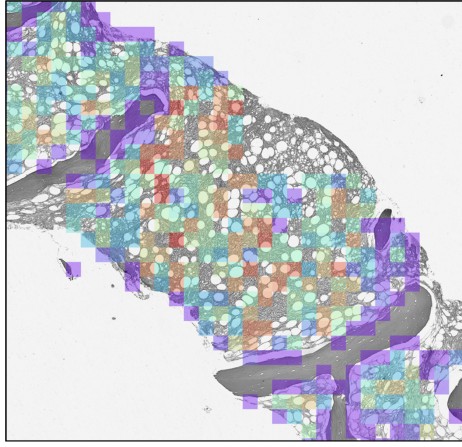

Figure 2: BM core with overlaid high/low model attention shown in warm/cool colors.

## Acknowledgments

We thank ARUP Laboratories for enabling this study, especially the R&I Digital Imaging Center for data and slide digitization support.

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
