# OpenReview forum: "Bone Marrow Fibrosis Grading Using Attention-Based Multiple Instance Learning"
_MIDL.io/2025/Short_Papers — MIDL 2025 - Short Papers_

### Official Review · Reviewer_gyAM · 2025-04-28

**Rating:** 3
**Confidence:** 4

**Summary:**

This paper proposes using Attention-Based Multiple Instance Learning (ABMIL) with DINO v2 features for bone marrow fibrosis grading. The authors employ a large language model (LLM) to extract slide-level labels from biopsy reports and validate their method on an internal dataset.

**Strengths:**

•	The experiments involve comparisons with modern pathology foundation models such as Virchow2, UNI, and GigaPath, which enhances the paper's relevance.
•	The logical flow of the methodology is sound.

**Weaknesses:**

•	The evaluation is limited to RMSE. In grading tasks, metrics like AUC and F1-score are also important and should be reported.
•	Figure 2 lacks ground truth for comparison, reducing its interpretability and impact.

---

### Decision · Program_Chairs · 2025-05-01

Accept